# From Wasteland to Bloom: Exploring the Organizational Profiles of Occupational Health and Well-Being Strategies and Their Effects on Employees’ Health and Well-Being

**DOI:** 10.3390/ijerph21081008

**Published:** 2024-07-31

**Authors:** Marie-Ève Beauchamp Legault, Denis Chênevert

**Affiliations:** Human Resources Management Department, HEC Montréal, 3000 Côte-Sainte-Catherine, Montréal, QC H3T 2A7, Canada

**Keywords:** occupational health and well-being, strategic management, latent profile analysis, MANOVA

## Abstract

Based on the signaling and conservation of resources theories, this study aims to identify different strategic organizational profiles related to occupational health and well-being (OHWB). Additionally, this study explores how these various organizational profiles impact employees’ well-being, specifically in relation to absenteeism, emotional exhaustion, work overload, intention to quit, and job satisfaction. Data were collected from 59 organizations and 2828 employees. The first phase of this study presents the latent profile analysis carried out to identify OHWB organizational profiles. This analysis reveals four organizational profiles that are metaphorically named according to the growth stages of plants (i.e., wasteland, sprouting, budding, and blooming OHWB profiles). The second phase of this study investigates the associations between the latent profiles assigned to the organizations with absenteeism, intention to quit, emotional exhaustion, feelings of work overload, and job satisfaction among their employees using MANOVA. The results show that organizational profiles influence employees’ health and well-being. Employees working in organizations with a low OHWB profile, known as the “wasteland profile”, tend to report more days of absenteeism, higher levels of emotional exhaustion, greater work overload, and lower job satisfaction. Employees are also more likely to express a greater intention to quit their jobs than those working in organizations with a higher OHWB profile (a “blooming profile”). This study is useful for organizations and practitioners seeking to understand how investing in a health and well-being strategy can benefit their employees.

## 1. Introduction

Organizations have a crucial role to play in promoting the well-being of their employees, as people spend a significant portion of their time at work [1]. However, occupational health issues can adversely affect absenteeism and productivity, thereby impacting the financial health of organizations. According to Mercer, the cost of lost productivity due to absenteeism alone amounts to CAD 16.6 billion annually for Canadian organizations [2], while disability caused by physical or mental health disorders affects one in five Canadians. Exacerbated by the COVID-19 pandemic, mental health issues are now a significant workforce challenge for organizations worldwide [3,4]. Despite the increased focus on employees’ health and well-being, many Canadian organizations are still reluctant to invest in health promotion [5]. This lack of enthusiasm may be due to Canadian employers contributing little to their employees’ healthcare costs, which could be attributed to Canada’s public healthcare system [6]. However, a survey reveals that 77% of Canadian professionals would be willing to leave their current jobs for an employer that prioritized their well-being [7]. This is an alarming statistic for organizations already struggling with a labor shortage. 

Organizations may need help navigating this complex landscape when implementing a workplace health and well-being strategy. Indeed, empirical studies on the effectiveness of practices or programs often yield contradictory or inconclusive results [8,9,10]. Given this ambiguity, it is reasonable to question whether contextual or strategic elements at the organizational level can impact employees’ health and well-being. 

Using the signaling theory [11] and the conservation of resources theory (COR) [12], this study focuses on the effect of organizational context and strategy in occupational health and well-being (OHWB) on employees’ health and well-being. We support the idea that it is possible to identify organizational strategic profiles regarding occupational health and well-being, and that strategic positioning sends a signal about the availability of resources to employees. For example, organizations with a strategic position that is more favorable to health and well-being would benefit employees’ health, unlike organizations that do not adopt this strategic position. Therefore, this exploratory study seeks to identify occupational health and well-being strategies by conducting a latent profile analysis. A better understanding of the organizational profiles linked to health and well-being at work can lead to better management and targeted interventions for employee well-being. This study also examines how these profiles interact and affect employees’ health.

In terms of contributions, this study demonstrates that it is possible to identify organizational profiles regarding occupational health and well-being strategies that play a differentiated role in employees’ health. This study also contributes to the signaling [11] and the conservation of resources theories [12] by demonstrating how signals and resources generated by an OHWB strategy can affect employees’ well-being. This study also stands out methodologically in that it uses latent profile analysis at the organizational level, whereas most studies typically use this methodology at the individual level [13,14,15,16]. Finally, this study emphasizes the significance of organizational commitment to OHWB for employees’ well-being and provides a benchmark with which organizations can evaluate their strategic positioning on OHWB.

The following sections present the theoretical framework and the development of the research hypotheses. First, the methodology and main results are presented in brief for each phase of this research (Phase 1: setting up organizational profiles; Phase 2: testing the organizational profiles on variables related to employee health and well-being). The discussion is then presented, highlighting this study’s theoretical and practical implications and limitations and identifying promising avenues for future research.

## 2. Theoretical Framework and Development of Hypotheses

Given the strategic role of human capital in the competitive positioning of organizations [17,18,19,20], we have seen the emergence, in recent decades, of the development of human resource (HR) strategies promoting employee health and well-being [21,22]. Indeed, changes in the nature and context of work demand more significant attention be paid to employee well-being. Therefore, organizational strategies and a fortiori HR strategy must now take an approach that will enhance employees’ well-being [23]. More specifically, the occupational health and well-being strategy that organizations advocate should reflect their strategic positioning regarding their human capital. OHWB’s strategic positioning is reflected in how an organization implements occupational health and well-being initiatives, including how management and the organization support, promote, communicate, determine, measure, evaluate, and adjust these initiatives. 

Consequently, the first hypothesis of this study is as follows: 

**Hypothesis** **1.***Organizations have different strategic profiles regarding OHWB*. 

Next, organizations’ strategic positioning regarding OHWB could affect employees’ well-being. Signaling theory [11] and COR theory [12,24] can help to understand this phenomenon. Signaling theory [11,25,26] states that a signaler (i.e., an organization) seeks to transmit signals to a receiver (i.e., employees) to influence their behaviors. Depending on the adopted OHWB strategy, the signals indicate the importance that the organization attaches to the health and well-being of its employees. An organization that prioritizes occupational health and well-being sends a different signal from one that does not. Signaling theory also proposes that the way in which receivers interpret signals affects their future attitudes and behaviors [26,27]. In the same way, the signal given out by an organization with a strategy that is strongly focused on occupational health is perceived positively by employees, which is then positively linked to their health. 

COR theory [12,24] assumes that individuals strive to obtain, preserve, nurture, and protect the things they value. Stress arises when employees feel that their resources are threatened or lost. Moreover, as employees try to protect, preserve, and increase these resources, gaining or maintaining them benefits their health and well-being [12,24]. An organization that prioritizes the well-being of its employees has a better chance of creating valuable resources and providing more resources to its employees than an organization that does not prioritize employees’ well-being. Therefore, the second hypothesis is as follows: 

**Hypothesis** **2.***OHWB strategies affect employees’ health and well-being differently. The more developed an organization’s health and well-being strategy is, the greater the positive impact on its employees*.

## 3. Methods

Data for this study were collected from organizations (n = 59) and employees (n = 2878) between December 2019 and June 2022. The university’s Research Ethics Board approved this study, and all participants (organizational respondents and employees) gave informed consent before completing the questionnaires. 

This section successively presents the methodology of (1) the latent profile analysis (n = 59 organizations) and (2) the exploration of the effect of these profiles on employees’ health and well-being (n = 32 organizations and 2828 employees).

### 3.1. Phase 1: Latent Profile Analysis

#### 3.1.1. Participants and Study Design

In total, 59 organizations in Quebec (Canada) completed a questionnaire intended for their HR manager between December 2019 and June 2022. Organizations were invited to participate in a research project on occupational health and well-being. We used social media, mainly LinkedIn, to distribute the call for participation. We also obtained a list of company names and emails from executives in Quebec and used it to send about a hundred emails and make solicitation calls. A total of 59 organizations responded positively to the call. This part of the study was conducted in French. Additionally, the questionnaire was pretested with ten individuals to ensure that all questions were understood clearly.

This study included organizations from 17 sectors (public administration, public services, retail, education, mining, finance and insurance, etc.), with 90% belonging to the service sector. Regarding size, 29% of the participating organizations have 1 to 49 employees, 29% have 50 to 249 employees, and 45% have more than 250 employees.

#### 3.1.2. Measures

Eight indicators were selected for the latent profile analysis. These indicators were adapted from the Worksite Health Promotion Capacity Instrument, a validated scale developed to assess the effectiveness of worksite health promotion by Jung et al. [28] (α = 0.92). We adapted the wording of the items to better match the subject of our study. The term “health” was changed to “occupational health and well-being” to be more representative of our hypotheses. A literature review and six case studies were also conducted to better understand the success factors involved in implementing an occupational health and well-being strategy and to confirm the choice of these indicators [29]. 

For each indicator, organizational respondents were asked to indicate on a seven-point Likert scale the extent to which they “strongly disagree” to “strongly agree” with the proposed indicators. Table 1 shows the indicators used for the latent profile analysis. 

### 3.2. Phase 2: Analyzing the Effects of OHWB Profiles on Employees’ Health and Well-Being

#### 3.2.1. Participants and Procedures

Of the 59 organizations comprising the latent profiles, 32 surveyed their employees to assess their health and well-being status, as well as their usage and perception of OHWB practices. 

Employees from all 32 of these organizations were invited to complete a survey. A total of 6522 employees opened the questionnaire. Of these 6522 employees, 3017 consented to participate and completed the questionnaire. Of these respondents, 189 questionnaires were eliminated due to missing or abnormal values. In all, the final sample comprised 2828 employees. 

Regarding descriptive statistics, 73% of the study participants identified as women. The average age of the participants was 44. Of all the respondents, 28% held a college diploma, 37% had a bachelor’s degree, and 15% had a master’s degree or MBA. Furthermore, 89% of the respondents worked full-time. Lastly, 48% of the participants had an organizational seniority of over ten years when completing the questionnaire.

#### 3.2.2. Measures

This portion of the study was carried out in both French and English. All instruments used in French were translated from English using a standard translation procedure [30]. A professional translator proofread the English questionnaire. The questionnaire also underwent pretesting with 10 respondents to ensure that all questions were comprehensible. To evaluate the effects of the OHWB organizational strategic profiles, we examined their association with absenteeism, intention to quit, emotional exhaustion, feelings of work overload, and job satisfaction.

##### Absenteeism

To measure absenteeism, employees were asked to indicate the number of days on which they had been absent from work during the last three months.

##### Intention to Quit

The intention to quit was measured using the two items on the scale developed by Bentein et al. [31]. On a Likert scale ranging from 1 = strongly disagree to 7 = strongly agree, employees were asked to comment on the following items: “I frequently think about leaving my job” and “It is very likely that I will look for a job in the next year”. The internal consistency of this instrument was α = 0.84. 

##### Emotional Exhaustion

Items relating to emotional exhaustion were collected using the emotional exhaustion subscale of the Maslach Burnout Inventory-Human Services Survey (MBI-HSS) [32]. This subscale comprises 7 items on a 7-point Likert scale ranging from never to always (e.g., “I feel exhausted at the end of a working day”, “I feel tired when I get up in the morning and have to face a new working day”, etc.). Regarding the psychometric properties of this scale, the internal consistency of this instrument was α = 0.90. 

##### Work Overload

The feeling of work overload was measured using two items of the three-item scale developed by Beehr et al. [33]. On a 7-point Likert scale, employees were asked to indicate their agreement (1 = strongly disagree, 7 = strongly agree) with the following statements: “My workload is such that I can’t take a break” and “There’s too much work to do in my job for everything to be done well”. The internal consistency of this instrument was α = 0.82.

##### Job Satisfaction 

To measure job satisfaction, participants were asked to indicate, on a scale from 0 to 100, the perceived percentage of satisfaction with their current job. 

Table 2 presents the descriptive and correlational statistics for these variables.

## 4. Results

### 4.1. Phase 1: Latent Profile Analysis and Identifying OHWB Strategic Profiles

To investigate our first research hypothesis, we conducted a latent profile analysis. This statistical analysis technique has recently attracted increased interest in management sciences [34]. This method of analysis mainly aims to identify different subgroups within a diverse population based on common characteristics [35,36].

The latent profile analysis was performed using SPSS (version 28.0.1.1) and R (version 2023.03.01+446) using the TidyLPA package and following the approach proposed by Bauer [37], Spurk et al. [34], and Wardenaar [38]. Bayesian information criterion (BIC) and Akaike information criterion (AIC), entropy, and the bootstrap likelihood ratio test (BLRT) were used to test model fit [34,37,39]. For the BIC and AIC, lower values indicate better fit [37]. For the BLRT, Bauer [37] recommends choosing the model before the BLRT becomes non-significant. For entropy, the score ranges between 0 and 1; values greater than or equal to 0.80 are desirable and the score must be at least greater than or equal to 0.60 [37].

Solutions for latent profiles ranging from 1 to 6 profiles were studied for the 59 participating organizations (see Table 3). The lowest BIC corresponds to the four-profile model. The lowest AIC corresponds to the six-profile model. It is recommended that the BIC index be prioritized over the AIC when choosing the number of profiles for better model selection [40]. Entropy measures the separation between different latent profiles. The higher the entropy, the better the class separation. The entropy for the four-profile model is 0.95, which is satisfactory. After adding a fifth profile, the BLRT is no longer significant (*p* = 0.13), supporting the selection of the four-profile model (*p* = 0.01). Based on interpretability, we decided to keep the four-profile model. These four profiles are illustrated in Figure 1. 

Profile 4 corresponds to 28.8% of the sample (n = 17) (see Table 4). This profile refers to organizations with a strong strategy focused on occupational health and well-being. Profile 3 represents 33.9% (n = 20) of the sample and includes organizations with a moderate level of development of their OHWB strategic position. Profile 2 comprises 22% of the sample (n = 13) and refers to organizations with an underdeveloped occupational health and well-being strategy. Finally, profile 1 comprises 15.3% of the organizations in our sample (n = 9). The organizations in this profile have the least developed strategy.

Each profile was named using a metaphor, to better illustrate its characteristics (as shown in Table 5). Profile 1 was named the “wasteland” profile and represents the organizations with a low OHWB strategic profile. Organizations with this profile have not yet invested in a workplace health and well-being strategy, much like uncultivated or unused land. This profile is characterized by a lack of OHWB initiatives. Senior management in these organizations provides minimal support for occupational health and well-being initiatives and rarely discusses or communicates the topic within the organization. Organizations in this category do not identify their needs or set quantifiable OHWB objectives and do not evaluate and adjust their practices accordingly.

Profile 2 is called the “sprouting” profile and represents a moderate OHWB strategy. As the name implies, this profile represents the initial stage of OHWB profile development, similar to a sprouting plant. Organizations with this profile have implemented a few OHWB initiatives with moderate levels of senior management support. There are frequent discussions and communications about the initiatives among employees. However, organizations in this profile do not identify employees’ needs or set quantifiable objectives, nor do they evaluate and adapt their practices to improve the OHWB strategy.

Profile 3 is called the “budding” profile, indicating a sustained OHWB strategy. As budding is a crucial stage in plant growth, this profile represents an important step in implementing the OHWB strategy. At this stage, organizations have already implemented a few OHWB initiatives with moderate support from management and have moderately frequent communication and discussion about these initiatives. What sets the budding profile apart from the sprouting profile is that organizations start showing interest in the results of their OHWB programs. They also take an interest in employees’ needs and develop quantifiable objectives for their programs. The evaluation and adjustment of practices are also part of the strategy at this level.

Profile 4 is the “blooming” profile, representing a strong OHWB strategy. At this stage, organizations have a blossoming occupational health and well-being culture. They have implemented significant OHWB initiatives and have the support of the management. Discussions and communications about health and well-being are frequent. These organizations determine their employees’ OHWB needs, set quantifiable objectives, evaluate them, and make the necessary adjustments. Just as a plant blooms when it is mature enough and the environmental conditions are right, during this stage, the organization puts everything in place to ensure health and well-being in the workplace.

### 4.2. Phase 2: Assessing the Effects of OHWB Profiles on Employees’ Health and Well-Being

After conducting the latent profile analysis with 59 organizations, a multivariate analysis of variance (MANOVA) was performed on 2828 employees (from 32 organizations) to test the hypothesis that there would be differences in means between absenteeism, intention to quit, emotional exhaustion, feelings of work overload, and job satisfaction, according to their OHWB profile. 

Out of the 32 organizations involved in this phase of the research, 12 organizations fit into Profile 4 (blooming profile), 10 organizations fit into Profile 3 (budding profile), 5 organizations fit into Profile 2 (sprouting profile), and 5 organizations fit into Profile 1 (wasteland profile). Table 6 illustrates the distribution of organizations and employees based on their OHWB’s strategic profile.

We associated each participating employee with the OHWB profile corresponding to their organization. In our sample, 1203 employees worked in a Profile 4 organization (blooming profile), 105 employees worked in a Profile 3 organization (budding profile), 317 employees worked in a Profile 2 organization (sprouting profile), and 1203 employees worked in a Profile 1 organization (wasteland profile). 

Table 7 includes descriptive statistics for the dependent variables for each organizational profile.

The MANOVA results showed that there was a significant difference between the organizational profiles and dependent variables (Pillai’s Trace: 0.162, F(21, 8118) = 22.10, *p* < 0.001, n^2^ = 0.054, observed power = 1.000). Follow-up ANOVAs were then carried out.

For the follow-up ANOVAs, we tested for the homogeneity of variances, and Levine’s test proved significant for all variables (*p* < 0.001). The results of the analysis of variance revealed significant differences for all the variables analyzed (*p* < 0.001) (see Table 8). Games–Howell post hoc significant difference tests were used to determine the differences between groups more accurately, as the unequal variance hypothesis was confirmed.

According to Figure 2, employees working in organizations with a low occupational health and well-being (OHWB) strategy (Profile 1) reported significantly more days of absence (M = 1.41, SD = 2.6, *p* < 0.001) compared to those working in organizations with other profiles. Specifically, employees in Profile 2 reported an average of 0.73 days of absence (SD = 2.04), employees in Profile 3 reported an average of 0.84 days of absence (SD = 1.82), and employees in Profile 4 reported an average of 0.85 days of absence (SD = 1.95).

We found a significant difference (*p* < 0.001) in the intention to quit (Figure 3) between employees from Profile 4 organizations (strong OHWB strategy) and those from Profile 1 organizations (low OHWB strategy). Employees from Profile 1 organizations (M = 2.73, SD = 1.98) are likelier to quit their organizations than Profile 4 employees (M = 2.07, SD = 1.52). Additionally, a significant difference (*p* < 0.01) between Profile 1 and Profile 2 (moderate OHWB strategy) (M = 2.31, SD = 1.70) was identified, with employees from Profile 1 organizations intending to quit the organization more than employees from Profile 2 organizations.

Employees in organizations with low OHWB strategies (Profile 1) reported higher levels of emotional exhaustion (M = 3.33, SD = 1.47) than employees in the other profiles (Figure 4). Specifically, employees in Profile 2 organizations with moderate OHWB strategies reported higher levels of exhaustion (M = 2.64, SD = 1.27) than those in Profile 4 organizations with strong OHWB strategies (M = 2.41, SD = 1.23). These differences are statistically significant (*p* < 0.001).

For work overload (Figure 5), employees working in Profile 1 organizations (low OHWB strategy) (M = 3.2, SD = 1.21) experienced significantly higher levels of work overload (*p* < 0.001) than those in Profiles 2, 3, and 4 organizations (Profile 2: M = 2.62, SD = 1.21; Profile 3: M = 2.37, SD = 1.14; Profile 4: M = 2.50, SD = 1.11).

Finally, there was a significant difference in job satisfaction (Figure 6) (*p* < 0.001) between Profile 1 (low OHWB strategy) (M = 71.76, SD = 19.27) and Profiles 2, 3, and 4 (Profile 2: M = 77.97; SD = 16.77; Profile 3: M = 81.81; SD = 12.24; Profile 4: 82.97, SD = 14.04). Employees in Profile 1 organizations were significantly less satisfied with their jobs than employees working in organizations classified as Profiles 2, 3, and 4. Furthermore, a significant difference (*p* < 0.001) was observed between Profile 2 (moderate OHWB strategy) and Profile 4 (strong OHWB strategy). Employees in Profile 2 organizations were generally less satisfied than those in Profile 4 organizations (Profile 2: M = 77.97; SD = 16.77; Profile 4: M = 82.97, SD = 14.04).

## 5. Discussion

The aims of this study were twofold: (1) to determine whether it is possible to identify different organizational profiles linked to occupational health and well-being, and (2) to analyze the effects of these profiles on the health and well-being of employees. Four strategic organizational profiles emerged from the latent profile analysis. These profiles are successively named “wasteland” (Profile 1), which characterizes organizations with a low strategic interest in OHWB; “sprouting” (Profile 2), which characterizes organizations with a moderate OHWB strategy; “budding” (Profile 3), referring to organizations with a sustained OHWB strategy; and “blooming” (Profile 4), which refers to organizations characterized by a strong OHWB strategy. 

The multivariate analysis of variance (MANOVA) revealed that organizational profiles are significantly associated with variables affecting employees’ health and well-being. This study shows that employees working in an organization that does not invest in occupational health and well-being (the “wasteland” profile) reported a higher rate of absenteeism (+93%), a higher intention to quit (+19%), more emotional exhaustion (+34%), a higher sense of work overload (+43%), and lower job satisfaction (−12.7%) than those working in Profile 4 organizations (the “blooming” profile). These findings can have a significant financial impact on organizations [41,42].

Moreover, this research indicates that, for most profiles, in organizations with more thoroughly developed OHWBs, absenteeism, intention to quit, and emotional exhaustion tend to decrease, and job satisfaction tends to increase. This means that the more resources an organization invests in an OHWB strategy and the more signals it sends to its employees about the importance it attaches to their health and well-being, the greater the positive impact on employees.

Studies show that employees’ health and well-being are critical determinants of their productivity and, indirectly, of organizational performance [43,44,45]. Nevertheless, not all organizations invest resources in addressing employees’ health issues, particularly in implementing workplace health and well-being initiatives or programs. Too often, employers fail to recognize that employees’ health constitutes one of their social obligations [6]. 

Just as fertile soil is necessary for plants to grow, organizations that invest resources in promoting the health and well-being of their employees are likely to feel the positive effects. This supports the idea that the level of development of an occupational health and well-being strategy can influence employees’ well-being, which aligns with the conservation of resources and signaling theories. Nevertheless, organizations should consider their culture and values when deciding on an occupational health and well-being strategy [19,46,47]. Some organizations may not develop their occupational health and well-being strategy based on context. For example, deciding not to cultivate land in agriculture may be a strategic decision, just as an organization may opt for a low occupational health and well-being strategy. Based on the concept of contingency [19], organizations must align their strategic level with their current circumstances [48]. However, our research indicates that organizations can improve their occupational health and well-being by prioritizing it through organizational strategies. Furthermore, this study suggests that employees’ health and well-being improve when their organization focuses on health and well-being.

### 5.1. Theoretical and Practical Implications

Theoretically, this study shows that investing in a workplace health and well-being strategy positively affects employees’ health and well-being. According to the COR and signaling theories [11,12,24], a robust OHWB strategy provides more resources and sends a stronger signal to employees than a weaker OHWB strategy. Additionally, in terms of methodology, this study innovates by using latent profile analysis at the organizational level. In practical terms, this study shows how organizational involvement and the development of an OHWB strategy are essential for employees’ well-being. It emphasizes the importance of effective interventions in promoting occupational health and well within organizations. The results of the latent profile analysis can serve as a useful guide for organizations seeking to enhance their OHWB strategy. By gaining a deeper understanding of the various organizational profiles, organizations can more effectively manage and intervene in the health and well-being of their employees.

### 5.2. Limitations

This study has some limitations that should be taken into consideration. First, the number of participating organizations for the latent profile analysis is limited. Using small samples for this analysis may lead to some issues with the class indicators and suitability indices [34]. Our statistical analyses did not reveal any of these concerns. However, conducting the same study with a larger sample of organizations may allow us to refine our results or identify additional profiles. It is important to note that the data used to analyze latent profiles of organizations and their impact on employees are cross-sectional; they therefore only provide a static view. A longitudinal perspective is required to better understand the dynamics of the profiles over time. It should also be noted that the organizations and employees who participated in this study were limited to a single Canadian province, which limits the generalizability of the results. A study mobilizing pan-Canadian data or conducting an international comparison of the different organizational profiles would be particularly instructive. Additionally, all the data collected at the organizational and individual levels were self-reported, increasing the risk of social desirability in responses. Future studies could use more objective data to overcome this limitation. 

## 6. Conclusions

In conclusion, this study responds to the need to better understand how an occupational health and well-being strategy can impact employees’ health. Among other things, this study demonstrates the existence of four organizational profiles in terms of occupational health and well-being, which we have succinctly named the “wasteland”, “sprouting”, “budding”, and “blooming” strategies. This study also shows that these strategies are associated with employees’ health and well-being. More specifically, employees working in an organization with a low OHWB strategy (the “wasteland” profile) reported more days of absenteeism, increased intention to quit, and a higher rate of emotional exhaustion than the other profiles. Our results also show that, for most profiles, the more developed the OHWB strategy, the higher the job satisfaction and the lower the levels of absenteeism, work overload, intention to quit, and emotional exhaustion. This study is useful for organizations and practitioners seeking to understand how investing in a health and well-being strategy can benefit their employees.

## Figures and Tables

**Figure 1 ijerph-21-01008-f001:**
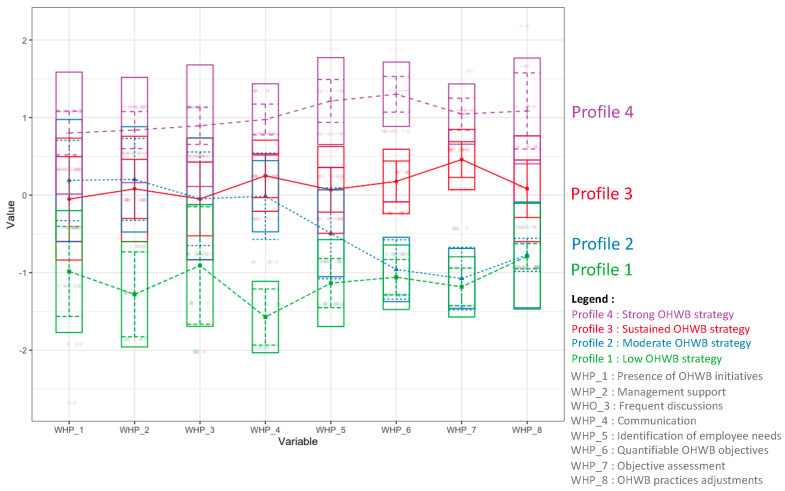
Graphic representation of the four latent profiles.

**Figure 2 ijerph-21-01008-f002:**
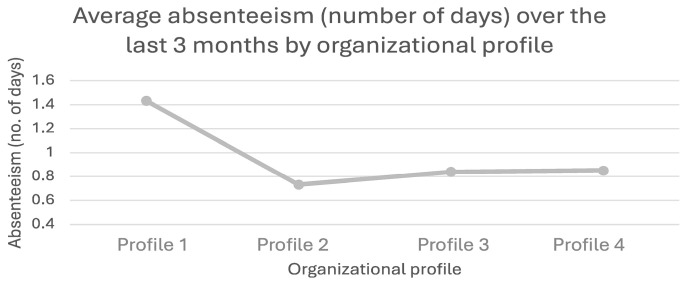
Employee absenteeism by organizational profile.

**Figure 3 ijerph-21-01008-f003:**
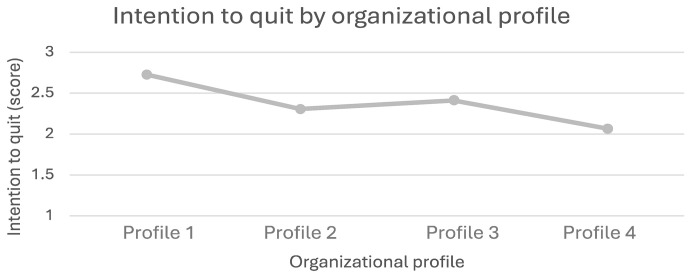
Intention to quit by organizational profile.

**Figure 4 ijerph-21-01008-f004:**
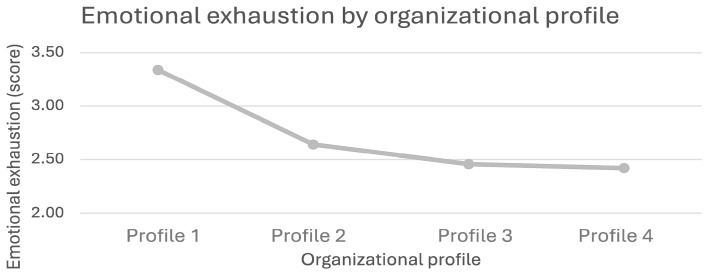
Emotional exhaustion by organizational profile.

**Figure 5 ijerph-21-01008-f005:**
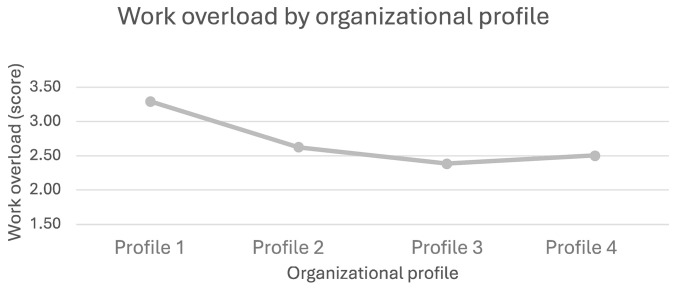
Sense of work overload by organizational profile.

**Figure 6 ijerph-21-01008-f006:**
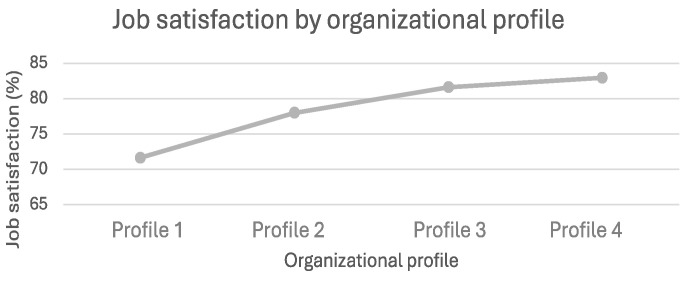
Job satisfaction by organizational profile.

**Table 1 ijerph-21-01008-t001:** Indicators used for the latent profile analysis.

1. The organization proposes initiatives or activities to improve the health and well-being of employees.
2. The organization’s management demonstrates a willingness to actively promote health and well-being in the workplace.
3. Employees’ occupational health and well-being are frequently discussed within the organization.
4. The organization effectively promotes occupational health and well-being within the organization and among employees.
5. The organization identifies employees’ occupational health and well-being needs before implementing its initiatives.
6. The organization defines quantifiable objectives before implementing its occupational health and well-being initiatives.
7. Occupational health and well-being initiatives are periodically evaluated.
8. Adjustments are made following evaluations of proposed OHWB initiatives

**Table 2 ijerph-21-01008-t002:** Descriptive statistics and correlations between variables.

Variables	M	SD	1	2	3	4	5
1. Absenteeism	1.09	2.29	1	0.167 **	0.268 **	0.091 **	−0.204 **
2. Intention to quit	4.62	3.26	0.167 **	1	0.477 **	0.241 **	−0.584 **
3. Emotional exhaustion	2.85	1.37	0.268 **	0.477 **	1	0.575 **	−0.566 **
4. Work overload	2.85	1.23	0.091 **	0.241 **	0.575 **	1	−0.388
5. Job satisfaction	77.18	17.67	−0.204 **	−0.584 **	−0.566 **	−0.338 **	1

** Correlation is significant at the 0.01 level.

**Table 3 ijerph-21-01008-t003:** Latent profile solutions.

No. of Profiles	AIC	BIC	Entropy	Prob_min	Prob_max	n_min	n_max	BLRT_*p*
1	1363.41	1396.65	1.00	1.00	1.00	1.00	1.00	
2	1165.84	1217.78	0.97	0.99	1.00	0.41	0.59	0.01
3	1102.05	1172.69	0.93	0.95	0.99	0.25	0.46	0.01
4	1052.23	1141.56	0.95	0.90	1.00	0.15	0.39	0.01
5	1050.63	1158.66	0.95	0.92	1.00	0.15	0.25	0.13
6	1049.58	1176.31	0.95	0.94	1.00	0.12	0.20	0.12

AIC, Akaike information criteria; BIC, Bayesian information criteria; BLRT, bootstrap likelihood ratio test.

**Table 4 ijerph-21-01008-t004:** Latent profiles and indicators.

	% org.	Presence of OHWB Initiatives (WHP_1)	Management Support (WHP_2)	Frequent Discussions (WHP_3)	Communication (WHP_4)	Identification of Employee Needs (WHP_5)	Quantifiable OHWB Objectives (WHP_6)	Objective Assessment (WHP_7)	OHWB Practices Adjustments (WHP_8)
Profile 4 (strong OHWB strategy)	28.8% (n = 17)	Strong	Strong	Strong	Strong	Strong	Strong	Strong	Strong
Profile 3 (sustained OHWB strategy)	33.9% (n = 20)	Moderate	Moderate	Moderate	Moderate	Moderate	Moderate	Moderate	Moderate
Profile 2 (moderate OHWB strategy)	22% (n = 13)	Moderate	Moderate	Moderate	Moderate	Low	Low	Low	Low
Profile 1 (low OHWB strategy)	15.3% (n = 9)	Low	Low	Low	Low	Low	Low	Low	Low

**Table 5 ijerph-21-01008-t005:** Four OHWB profiles.

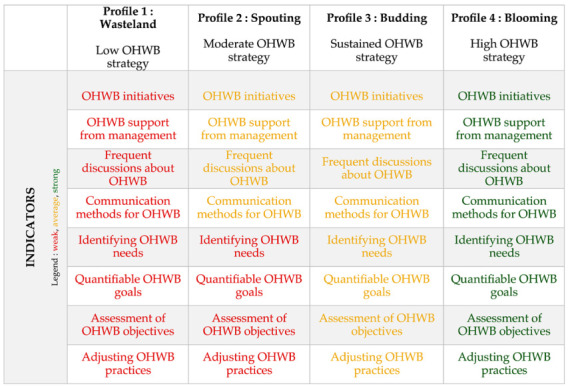

**Table 6 ijerph-21-01008-t006:** Distribution of participating organizations and employees according to their OHWB strategic profiles.

	No. of Organizations	No. of Employees
Profile 4 (strong OHWB strategy) Blooming profile	12 (37.5%)	1203 (42.5%)
Profile 3 (sustained OHWB strategy) Budding profile	10 (31.2%)	105 (3.7%)
Profile 2 (moderate OHWB strategy) Sprouting profile	5 (15.6%)	317 (11.2%)
Profile 1 (low OHWB strategy) Wasteland profile	5 (15.6%)	1203 (42.5%)

**Table 7 ijerph-21-01008-t007:** Descriptive statistics for dependent variables according to organizational profiles.

Descriptive Statistics				
	OHWB Organizational Profiles			
Dependent Variables	Profile 1	Profile 2	Profile 3	Profile 4
	Mean ± SD	Mean ± SD	Mean ± SD	Mean ± SD
Absenteeism	1.43 ± 2.64	0.73 ± 2.04	0.84 ± 1.82	0.85 ± 1.95
Intention to quit	2.73 ± 1.90	2.31 ± 1.70	2.41 ± 1.55	2.07 ± 1.50
Emotional exhaustion	3.34 ± 1.48	2.64 ± 1.28	2.46 ± 1.19	2.42 ± 1.13
Work overload	3.30 ± 1.22	2.62 ± 1.22	2.37 ± 1.14	2.50 ± 1.11
Job satisfaction	71.67 ± 19.27	77.97 ± 16.76	81.81 ± 12.25	82.97 ± 14.04

**Table 8 ijerph-21-01008-t008:** Results of ANOVAs.

Dependent Variables	Type III Sum of Squares	Df	Statistical Mean Square	F-Value	Sig.	Partial Eta-Square	Observed Power
Absenteeism	242.179	3	80.726	15.454	<0.001	0.017	1.000
Intention to quit	246.968	3	82.323	28.141	<0.001	0.031	1.000
Emotional exhaustion	503.210	3	167.737	97.682	<0.001	0.101	1.000
Work overload	398.145	3	132.715	96.356	<0.001	0.099	1.000
Job satisfaction	73,035.450	3	24,345.150	86.392	<0.001	0.090	1.000

## Data Availability

The data presented in this study are available on request from the corresponding author due to privacy reasons.

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
