# Peer review of "From Wasteland to Bloom: Exploring the Organizational Profiles of Occupational Health and Well-Being Strategies and Their Effects on Employees’ Health and Well-Being"

_ijerph, 2024, doi:10.3390/ijerph21081008_

Round 1

Reviewer 1 Report

Comments and Suggestions for Authors

The article is very interesting and up-to-date.

However, some work is needed to make it perceptible to other authors.

 The methodology should be described in such a way as to be reproducible by other authors. The origin of the tools used and their scientific validity should also be indicated.

 The methodology chapter should be separated from the results chapter and each of the variables should be described so that it is possible to critically analyse the results obtained.

The bibliography should be more robust, so that it is possible to critically analyse the state of the art on the subject.

The conclusions should be revised and completed, highlighting what the article presents as a new contribution to the scientific and technical community on the subject.

Comments on the Quality of English Language

Author Response

Please see the attachment PDF. Thank you ! 

Reviewer 2 Report

Comments and Suggestions for Authors

The authors conducted a cross-sectional study to categorize organizations with certain profiles and investigate the associations between certain these organizational profiles and occupational health and well-being. The study is interesting, relevant, and would be helpful to managers and occupational researchers.

 I have some comments and suggestions that I believe can improve the paper.

Overall, the paper needs extensive reorganization and some English language editing.

Abstract

Lines 16-18: This sentence is not necessary and can be removed.

Line 20: Recommend removing “Cross-sectional quantitative” from the beginning of this sentence.

Lines 25-28: Suggest rewriting this sentence in the active voice, e.g., “We investigated associations between the profile assigned to the organization with absenteeism, intention to quit,…..using MANOVA”.

Add a Conclusion to the end of the Abstract.

Introduction

The Introduction could be shortened by placing certain sections in the Discussion.

Page 2, lines 61-66: This paragraph seems more appropriate in the Discussion section.

Lines 80-91: This paragraph should be placed under a sub-heading “Limitations and Strengths” in the Discussion.

Page 3, lines 114-115: This first sentence seems like it belong under Hypothesis 2.

Methods and Results

This section seems confusing and needs major reorganization. I suggest making Methods and Results two major headings.

Under Methods, show these subheadings:

Participants and Study Design

Independent variables (or Exposures)

Dependent variables (or Outcomes)

Covariates

Statistical Analysis

Under Results, you don’t need to show subheadings. Write the results of Table 1 and/or Figure 1, Table 2 and/or Figure 2, etc., etc.

P. 4, lines 174-175: Revise sentence to read as “….was adapted to better match the subject….”.

Pp. 5, line 190: French words?

P. 5, lines 203-212: Do you think this paragraph or part of it belongs under Methods rather than Results?

P. 7, lines 239-249: These two paragraphs definitely belong under Methods.

P. 8, lines 251-260: These paragraphs belong under Results.

P. 8-10, lines 262-295, 298-303, 309-314: These descriptions of variables and analytical approach belong under Methods.

Table 6: Use 2 points after the decimal consistently in this table. Recommend calculating p-values showing how each profile differs with regards to the dependent variables.

Also, it would be easier for readers to follow the data in this table if it is rearranged as follows:

Profile 1

Profile 1

Profile 1

Profile 1

P-value

Mean ± SD

Mean ± SD

Mean ± SD

Mean ± SD

Absenteeism

1.43 ± 2.64

0.73 ± 2.04

Etc.

Table 7: The information presented is not clear. Where is the independent variable? What is this table meant to show?

Figures 2-6: Recommend labeling the vertical axis with the units, e.g., Absenteeism (no. of days). Were these data adjusted for any confounders?

Discussion

P. 12, lines 363-364: This sentence lists one of the objectives. Suggest including the other objective as a second sentence.

Lines 375-399: Place these paragraphs under Results and only write a summary here in the Discussion.

P. 14, lines 424-426: I disagree with this sentence (“However, our research indicates….”). Suggest revising or removing it.

P. 14, line 443: Suggest renaming this sub-heading as “Limitations and Strengths”. Place information regarding future research under Conclusion.

P. 14, line 463: Replace ‘strategies’ with ‘organizational profiles’. Since this is a cross-sectional study and not a longitudinal study, I suggest replacing ‘impact’ with ‘associated with’. For example, “…these different organizational profiles are associated with employee health…”.

Comments on the Quality of English Language

Minor English language editing is needed.

Round 2

Reviewer 1 Report

Comments and Suggestions for Authors

Thank you very much for the improvements to your article, it has improved significantly.

It just needs a final revision of minor errors.

Comments on the Quality of English Language

Thank you very much for the improvements to your article, it has improved significantly.

It just needs a final revision of minor errors.

Reviewer 2 Report

Comments and Suggestions for Authors

The authors have addressed all of my comments and suggestions to my satisfaction. The manuscript is now much improved. I have no additional comments.